# Sound Wave Absorption Coefficient and Sound Velocity in Thermally Modified Wood

Przemysław Mania [1],*, Artur Flach [2] and Marta Pilarska [1]

1 Department of Wood Science and Thermal Techniques, Faculty of Forestry and Wood Technology, Poznań University of Life Sciences, Wojska Polskiego 38/42, 60-627 Poznań, Poland

2 AGH University in Krakow, Al. Mickiewicza 30, 30-059 Kraków, Poland

* Correspondence: przemyslaw.mania@up.poznan.pl; Tel.: +48-618-487-448

**Abstract:** The present work analyses the absorption coefficient of sound waves and the speed of sound propagation in thermally modified wood. The high resistance to weathering, fungi, and better dimensional stability, and therefore the broad physical properties of this material, are well known. However, the literature lacks numerous analyses of its acoustic characteristics. During the study, high-density species, such as oak, red oak, and beech were used, in contrast to pine. Pine wood during this test was characterised by a most rapid increase in the sound absorption coefficient value, in the range of 1000–6300 Hz, and reached the highest value from all wood species. Among all species, the highest value of the examined parameter was obtained for beech wood and pine wood, which were 0.213 (at frequency 3 kHz) and 0.183 (at 6.3 kHz), respectively. The sound velocity decreased for all species only in the tangential direction.

**Keywords:** wood; sound absorption coefficient; sound velocity; dynamic modulus of elasticity; heat treatment

## 1. Introduction

Noise reduction on highways is achieved due to the installed acoustic screens, which act as sound absorbers or reflectors. These structures are most often made of concrete, aluminium, transparent polycarbonate, acrylic, or PVC screens. The aesthetics of such screens can be improved by using thermally modified wood, which is characterised by high dimensional stability and resistance to external factors. The concept of wood modification includes the interaction of various factors that improve wood properties or produce a new material that will not threaten the environment [1]. Thermal modification of wood is a well-known topic [2–6]. In addition to this commonly used method, one can also find others regarding changes in the properties of archaeological wood [7,8]. During this process, changes occur in the structural components of the wood. Cellulose, hemicellulose, and lignin are mainly degraded [9]. The decomposition of hemicelluloses reduces the amount of food for fungi, due to which thermally modified wood is more resistant to fungi and bacterial activity [10,11]. This type of process improves its dimensional stability [12] and mechanical properties [13–15]. Changes in these properties enable the use of wood on an even larger scale, due to which we also use this material in the exterior architecture. Despite much information available about thermally modified wood, little is known about its acoustic properties, which are the parameters needed to use this material as a sound absorber.

The porosity of the material promotes sound absorption. The sound wave absorption coefficient is a physical quantity that determines the ability of a material to absorb sound from the environment. This parameter takes a value from 0 to 1, where 1 is the total absorption of the energy applied to a given material and 0 is its total reflection. The ability of pine and oak wood to absorb sounds is influenced by their structure, intertwining early and late wood layers of different densities and porosities [16]. According to a study

conducted by Smardzewski and Kamisiński [17], there is a visible relationship between the absorption coefficient and wood density. The species used in the study were oak, ash, sapele, pine, balsa, birch, poplar, meranti, alder, and elm. An increase in the value of the sound absorption coefficient was noted with increasing density in the frequency ranges of 2 kHz, 1 kHz, and 4 kHz. Thermal modification of wood also leads to changes in its density [18]. According to the research, a more significant weight loss of spruce wood was recorded with an increase in the temperature of the modification process. At 180 °C, the decrease in density was 4.8%. The decomposition and evaporation of some chemical components of wood also decrease its hygroscopicity [19]. By examining the acoustic parameter of thermally modified wood, its potential as a sound-absorbing material can be determined. In the case of thermally modified wood, the study by Chung's [9] proves that an increase in temperature and the duration of the modification process increases the sound absorption coefficient. The growth rate of the coefficient was greater for the high-frequency bands. When the process was carried out at 220 °C for 18 h, the sound absorption coefficient for the frequency of 250 Hz was 0.028, whereas for 4 kHz, it was 0.037. Japanese larch wood was tested as well. Other scientists have also dealt with the sound absorption coefficient, e.g., in pellets [20], musical instruments [21], or other materials [20,22–24], and even with some modifications [25–27]. The sound absorption properties and the potential of wood and wood-based materials as sound absorbers were also characterised [28,29]. There are no reports on the effect of thermal modification on the sound absorption coefficient for other types of wood. The aim of this study was to determine the sound wave absorption coefficient of thermally modified wood. Research in this area was carried out on the wood of Scots pine, European beech, pedunculate oak, and red oak. In addition, how the sound velocity changed in the analysed wood after thermal modification was checked.

## 2. Materials and Methods

In order to study the sound wave absorption coefficient of thermally modified wood, four species were used as follows: scots pine (*Pinus sylvestris* L.), beech (*Fagus sylvatica* L.), oak (*Quercus robur* L.), and red oak (*Quercus rubra* L.). The tests were performed both before and after the thermal treatment.

The study of the sound wave absorption coefficient was carried out in two stages. The same material was compared before and after thermal modification to obtain the values of the tested features. The analysed wood species were tested using an impedance tube during the first stage. In the second stage, the twin material was thermally modified. Samples were subjected to thermal modification by the most often used method, in a steam atmosphere, according to the procedure of ThermoWood [2,3,18,30]. The wood modification was carried out under industrial conditions at 190–213 °C. After the thermal treatment, the samples were conditioned under laboratory conditions (T = 20 °C and RH = 65%). When the samples reached the same equilibrium moisture content as before the modification, they were subjected to testing for the sound wave absorption coefficient. Due to using an impedance tube, in particular two tubes, the samples were in the form of discs with diameters of 29 mm and 100 mm. The thickness of each sample was 23 mm. Figure 1 shows an exemplary pair of samples made of beech wood before and after the thermal modification.

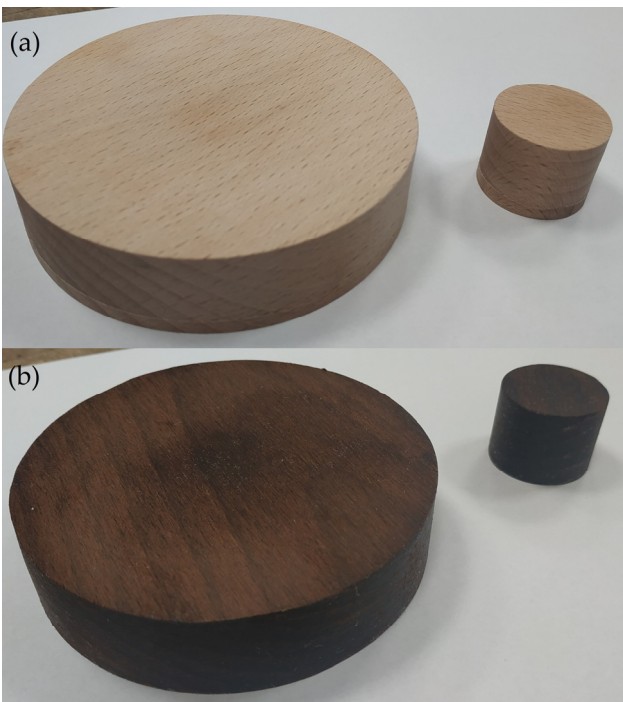

**Figure 1.** A pair of beech wood samples before (**a**) and after thermal modification (**b**).

The samples of unmodified and modified wood of a given species were obtained from a single board (Figure 2). The board from which the samples were cut out was 50 mm thick. After marking the places where the samples were to be cut, the board was cut lengthwise (along the dotted line). Then, one part was subjected to thermal modification, and the control samples were cut out from the other part. After modification, the modified wood samples were cut out from the previously marked places. Elements with different diameters were cut in pairs to obtain samples corresponding to each other in density. At least six pairs of samples were prepared for each species.

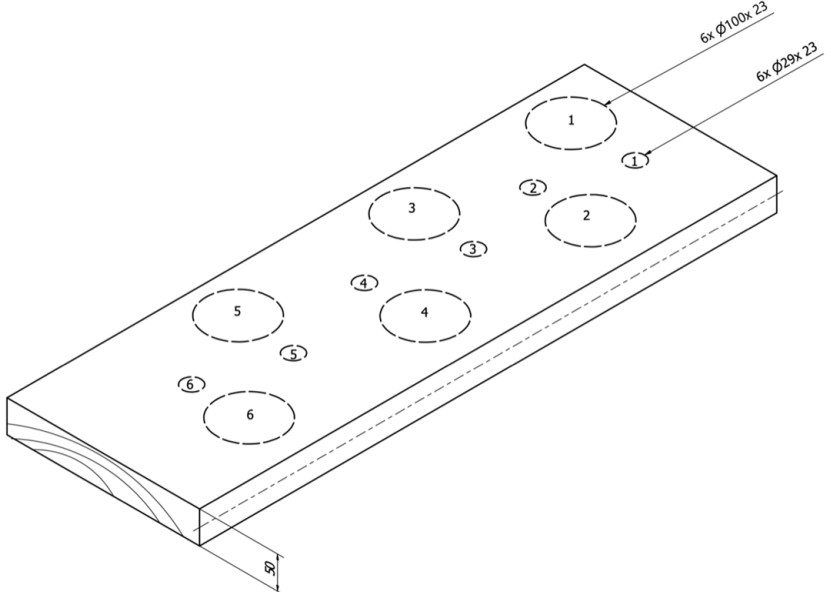

**Figure 2.** Sample cutting scheme.

The sound wave absorption coefficient was determined using an impedance tube (Bruel & Kjaer, type 4206) with a PULSE measurement card. This apparatus consisted of

an impedance tube, a loudspeaker, a microphone probe, and a generator (Figure 3). The measuring range was 50–1600 Hz for samples with a diameter of 29 mm and 500–6300 Hz for samples with a diameter of 100 mm. The test consisted of placing a sample in an impedance tube, onto which sound waves were generated perpendicular to its surface. The sound pressure, which determines the field interference distribution, was measured using two microphones. As a result, information on the reflection and absorption coefficient of the signal passing through the tested material was obtained. The tests were performed using the ISO 10534-2 standard [31].

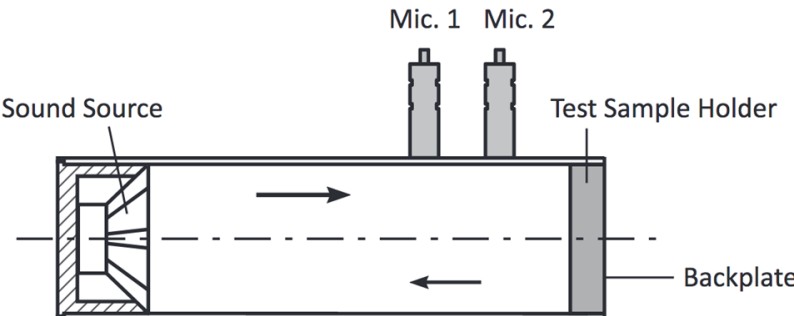

**Figure 3.** Scheme of the impedance tube (Bruel & Kjaer, type 4206).

The speed of sound propagation in the three-dimensional part was measured by measuring the time it took for the wave to travel through the samples. The material sampler type 543E (UNIPAN, Poznań, Poland) transmitted and received transducers with a frequency of 1 MHz. The speed of the sound wave on each sample was calculated using the following formula:

$$V = \frac{s^{L,R,T}}{t} \left( \text{m·s}^{-1} \right) \tag{1}$$

L, T, and R refer to the sample size in the longitudinal, tangential, and radial directions (m), respectively, with the propagation time (s). The sound absorption coefficient of the samples was determined. The sound velocity and the sound absorption coefficient were determined on the same samples.

Since the wood's speed of sound propagation and density were known, the sonic MOE ($MOE_D$), or dynamic $MOE_D$, was only calculated for the longitudinal direction using the following formula:

$$MOE_D = V^2 \cdot \rho \ (\text{MPa}) \tag{2}$$

All determinations were made on wood with the same moisture content. Control samples were seasoned at 20 °C and 40% air humidity. To obtain the same equilibrium moisture content of the modified samples, they were placed in a chamber at the same temperature with an air humidity of 65%. This allowed us to obtain the equilibrium moisture content of the wood at 7%.

The experimental data were analysed using the DellTMStatisticaTM13.3 software with the analysis of variance (ANOVA). Tukey's HSD test revealed significant differences between the mean values of the parameter describing the properties of untreated and heat-treated wood species. The comparison tests were performed at a 0.05 significance level. Identical superscripts, e.g., a, b, and c, denote no significant difference between the mean values of the investigated properties.

## 3. Results and Discussion

Table 1 shows the wood density of the material before and after thermal modification. The results are mean values calculated for at least six pairs of samples from each wood species.

**Table 1.** Wood density before and after heat treatment.

| Wood Species | Oven Dry | | |
|---|---|---|---|
| | $\rho$ | $\rho_{HT}$ | $\Delta\rho$ |
| | (kg·m$^{-3}$) | | (%) |
| Beech | 743 ± 32.6 | 579 ± 38.0 | −22.1 ± 1.02 |
| Oak | 686 ± 41.5 | 573 ± 46.1 | −16.5 ± 0.94 |
| Red oak | 679 ± 49.1 | 601 ± 50.7 | −11.5 ± 0.92 |
| Scots pine | 641 ± 43.5 | 564 ± 38.3 | −12.0 ± 0.98 |

Based on the above information, it can be seen that after the thermal modification process, the wood density value decreased. This decrease is caused by a loss of mass, which is related to the decomposition of some chemical components of the wood structure. The decrease in density depends on the temperature of the modification process; the higher the temperature, the higher the density loss. The decomposition and evaporation of some chemical components of wood decrease its hygroscopicity [18]. The most significant percentage density changes were recorded for the sample of beech wood, which was 23%, followed by oak at 16%, pine at 13%, and red oak at 12%.

The analysis of the sound propagation speed was divided according to the species used in the study and their dimensions. In Table 2, a summary of the average sound propagation speed values in the direction along the fibres and in the radial and tangential direction in the material before thermal modification is presented.

**Table 2.** Comparison of sound propagation velocity values before modification (V$_{RR}$—sound velocity in the radial direction; V$_{LL}$—sound velocity along the fibres; V$_{TT}$—sound velocity in the tangential direction).

| Wood Species | 29 mm | | | 100 mm | | | Average | | |
|---|---|---|---|---|---|---|---|---|---|
| | V$_{RR}$ | V$_{LL}$ | V$_{TT}$ | V$_{RR}$ | V$_{LL}$ | V$_{TT}$ | V$_{RR}$ | V$_{LL}$ | V$_{TT}$ |
| | [m/s] | | | | | | | | |
| Beech | 2439 | 5331 | 1457 | 2274 | 4831 | 1354 | 2356 | 5081 | 1405 |
| Oak | 2203 | 5458 | 1219 | 2115 | 5178 | 1355 | 2159 | 5318 | 1287 |
| Red oak | 2516 | 5247 | 1228 | 2025 | 4257 | 1268 | 2271 | 4752 | 1248 |
| Scots pine | 1865 | 6094 | 901 | 2172 | 6430 | 1060 | 2019 | 6262 | 981 |

From the data presented in Table 2, it can be confirmed that the highest sound propagation speed occurred in the longitudinal direction, followed by the radial and tangential directions, respectively. The relationship VL > VR > VT, as mentioned by Bucur [32], was also confirmed in this case. Due to the larger diameter of the 100 mm diameter samples, a slightly lower wave velocity was observed. The highest results were recorded for the pine samples in the direction along the fibres. The longest anatomical elements characterise this species. The average tracheids length in pine was approximately 3.1 mm, whereas the average length of the fibres was approximately 1 mm [33]. Then, considering the obtained average values, oak, beech, and red oak can be distinguished.

After the process of thermal modification of wood, the values of the sound propagation velocity changed. The results are presented in Table 3.

**Table 3.** Comparison of sound propagation velocity values after thermal treatment (V—sound velocity; R—radial direction; L—direction along the fibres; T—tangential direction).

| Wood Species | 29 mm | | | 100 mm | | | Average | | |
|---|---|---|---|---|---|---|---|---|---|
| | $V_{RR}$ | $V_{LL}$ | $V_{TT}$ | $V_{RR}$ | $V_{LL}$ | $V_{TT}$ | $V_{RR}$ | $V_{LL}$ | $V_{TT}$ |
| | | | | | [m/s] | | | | |
| Beech | 1951 | 5365 | 1066 | 1522 | 5708 | 1302 | 1737 | 5536 | 1184 |
| Oak | 2368 | 5283 | 1149 | 2345 | 5206 | 1321 | 2230 | 5244 | 1235 |
| Red oak | 2650 | 5455 | 1044 | 2230 | 5629 | 1266 | 2440 | 5542 | 1155 |
| Scots pine | 1820 | 6364 | 885 | 2062 | 6495 | 811 | 1991 | 6430 | 848 |

The tested values decreased for the 29 mm and 100 mm samples for all species, only in the tangential direction. In the radial direction, there was an increase in the tested speed for specimens with 29 mm oak and red oak by 7.5% and 5.3%, respectively, and for 100 mm oak and red oak by 10.5 and 10.1%, respectively (Table 3). In the case of the remaining samples, the value of the sound propagation velocity in this direction decreased. The reason for this phenomenon may be the presence of specific wood rays in the oaks. Both species have very wide tree rays. Their structure can help to increase the sound velocity in wood. In addition, oak wood was characterised by a relatively small loss of density.

Table 4 shows the material's dynamic modulus of elasticity values before (MOE_D) and after thermal modification (MOE_DHT). These are the values calculated for the average value of the sound speed propagation only in the longitudinal direction.

**Table 4.** Dynamic modulus of elasticity before and after heat treatment.

| Wood Species | Oven Dry | | |
|---|---|---|---|
| | $MOE_D$ | $MOE_{DHT}$ | $\Delta MOE_D$ |
| | (MPa) | | (%) |
| Beech | $19{,}182 \pm 1922$ | $17{,}745 \pm 2089$ | $-7.5 \pm 0.42$ |
| Oak | $19{,}401 \pm 1935$ | $15{,}757 \pm 2416$ | $-18.8 \pm 0.68$ |
| Red oak | $15{,}333 \pm 1757$ | $18{,}459 \pm 2332$ | $+20.4 \pm 1.72$ |
| Scots pine | $25{,}135 \pm 2134$ | $23{,}318 \pm 2688$ | $-7.2 \pm 0.21$ |

Generally, the modulus of elasticity determined using ultrasonic testing is much higher than that using destructive testing. The obtained MOE values, therefore, exceed Young's modulus values found in the literature [33]. As mentioned above, the density of all the analysed samples decreased. In the case of the sonic modulus of elasticity, a similar decrease was recorded in pine and beech wood, which was approximately 7.5%. However, the change in the MOE for oak wood was surprising. In pedunculate oak (*Quercus robur* L.), a reduction in MOE by more than 18% was noted, whereas, in red oak, it increased by more than 20%. This increase was mainly due to an increase in the speed of sound propagation in this species. For red oak, it was greater than 10%. By analysing the relationship from which the MOE was calculated, it can be seen that the sound velocity was squared. In addition, considering that the density decrease was the lowest of all the analysed species (Table 1), such a high increase in MOE was not surprising.

The sound absorption coefficient was interpreted for each species, and then the materials used before and after the modification process were compared. In order to better illustrate the variability of the wave absorption depending on the frequency, a relationship is presented in Figure 4.

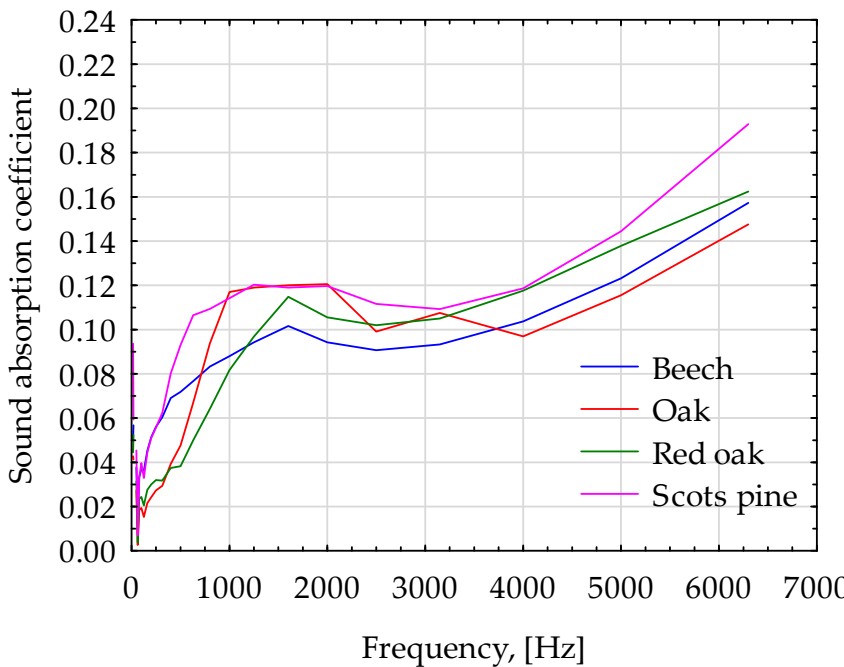

**Figure 4.** Sound absorption coefficient in beech, oak, red oak, and pine wood before thermal modification.

The human hearing range is from 16 Hz to 21,000 Hz, with the human ear being the most sensitive in the 2000–3000 Hz range [34]. The graph presented in Figure 2 shows that the sound absorption coefficient increases with increasing frequency. In the initial range of 60–500 Hz, scots pine wood achieves the highest results, whereas red oak wood achieves the lowest. This situation changes in the 500–2500 Hz range, whereas oak wood shows the highest sound absorption coefficient. At frequencies of 1600 Hz, it reaches a value of 0.12. In this range, an increase in the examined parameter for red oak wood is also visible. In the 500–3500 Hz frequency range, beech wood is characterised by the lowest value of the analysed parameter. Beechwood also shows the highest density at the beginning of the process. During this test, pine wood is characterised by a rapid increase in the sound absorption coefficient value, in the range of 1000–6300 Hz, and reaches the highest value of all wood species. The mean values for oak and red oak wood are very similar. The sound absorption coefficients of porous materials such as wood are proportional to the porosity and roughness of the inner pore wall. The differences in the sound absorption coefficients of both species are statistically insignificant. Statistically significant differences are observed between the remaining species. Similar results for the sound absorption coefficient were obtained by Smardzewski [16]. By analysing different wood species, he described that the parameter did not exceed 0.2 for a frequency of 2 kHz.

The thermally modified wood was subjected to the same test. The average sound absorption coefficient is shown in Figure 5. The figure shows a rapid increase in the tested parameter above 800 Hz for all the species. Among all species, the highest values of the examined parameter were obtained for beech wood and pine wood, which were 0.213 (at frequency 3 kHz) and 0.183 (at 6.3 kHz). The increase in the sound absorption coefficient due to heat treatment in the high-frequency band is due to the increased porosity. During heat treatment, wood density was reduced, and the intercellular layers were exposed due to the shrinkage of the cell wall. This increased the porosity of the wood. If we calculate the porosity (C) of wood using Formula (3)

$$C = \left(1 - \frac{\rho_D}{1.5}\right) \cdot 100 \ (\%), \tag{3}$$

where 1.5 is the density of the wood substance, and $\rho_D$ is the density of the wood, we obtain 50.5% for beech, 54.3% for oak, 54.7% for red oak, and 57.3% for pine wood. The thermal modification increased the porosity to 61.4%, 61.8%, 59.9%, and 62.4%, respectively. The increase in porosity was statistically significant for all the analysed wood species. The highest increase was obtained for beech wood, which was approximately 22%, and the lowest increase was for pine wood, which was 8.95%.

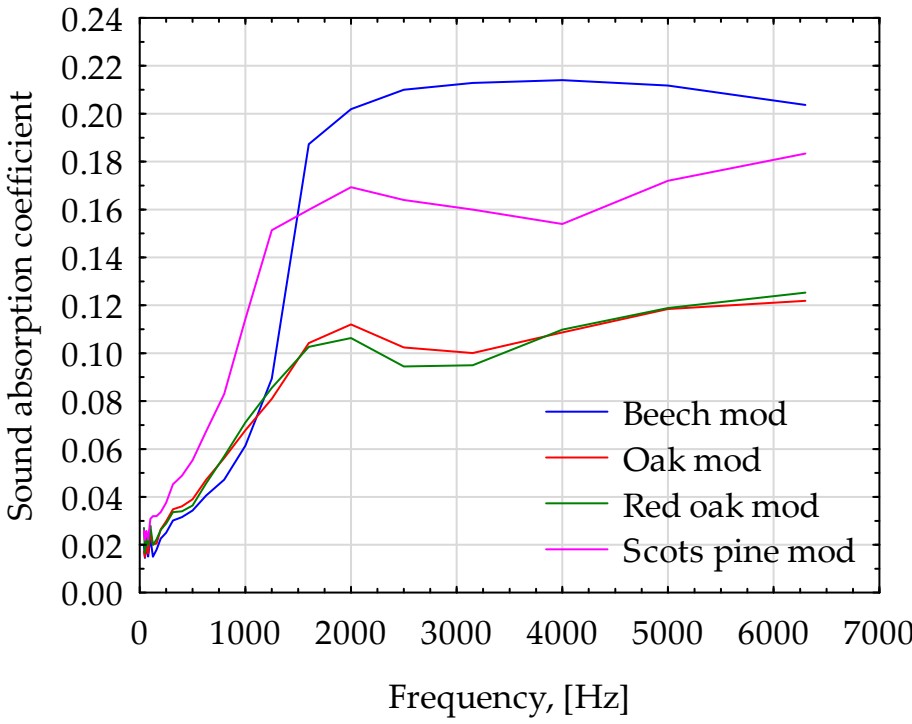

**Figure 5.** Sound absorption coefficient in beech, oak, red oak, and pine wood after thermal modification.

Based on Figure 6, it is possible to determine the impact of thermal modification on the value of the sound absorption coefficient in the wood of individual species. In the case of oak wood, the tested parameter obtained lower values in the frequency range of 500–2000 Hz after the thermal modification process. The most significant difference was noted at 1000 Hz, where the decrease was 43% compared to that of the control wood. In the case of red oak wood, the observed differences were minor. Beechwood, after modification, was characterised by a higher wave absorption coefficient than wood before thermal modification. The observed increase is visible for the frequency range 1500–5500 Hz, with the wave absorption coefficient value increasing by more than 100% for a frequency of 3000 Hz. Visible differentiation can be seen in the case of pine wood, where the sound absorption coefficient values also increased after the thermal modification of the wood. In the 800–4000 Hz frequency range, the increase was up to 25% compared to the results of the control wood. Therefore, the greatest increase in the analysed parameter was recorded in the wood with the lowest initial density (scots pine) and in the wood with the greatest density loss (beech wood).

The observed increase in the sound absorption coefficient was similar to that obtained by Kang [35]. In the study, the wood was subjected to a different modification process. After delignification, the wood obtained a 20–30% higher sound absorption coefficient than the control wood. The authors also explained that this change was due to increasing the porosity of the wood.

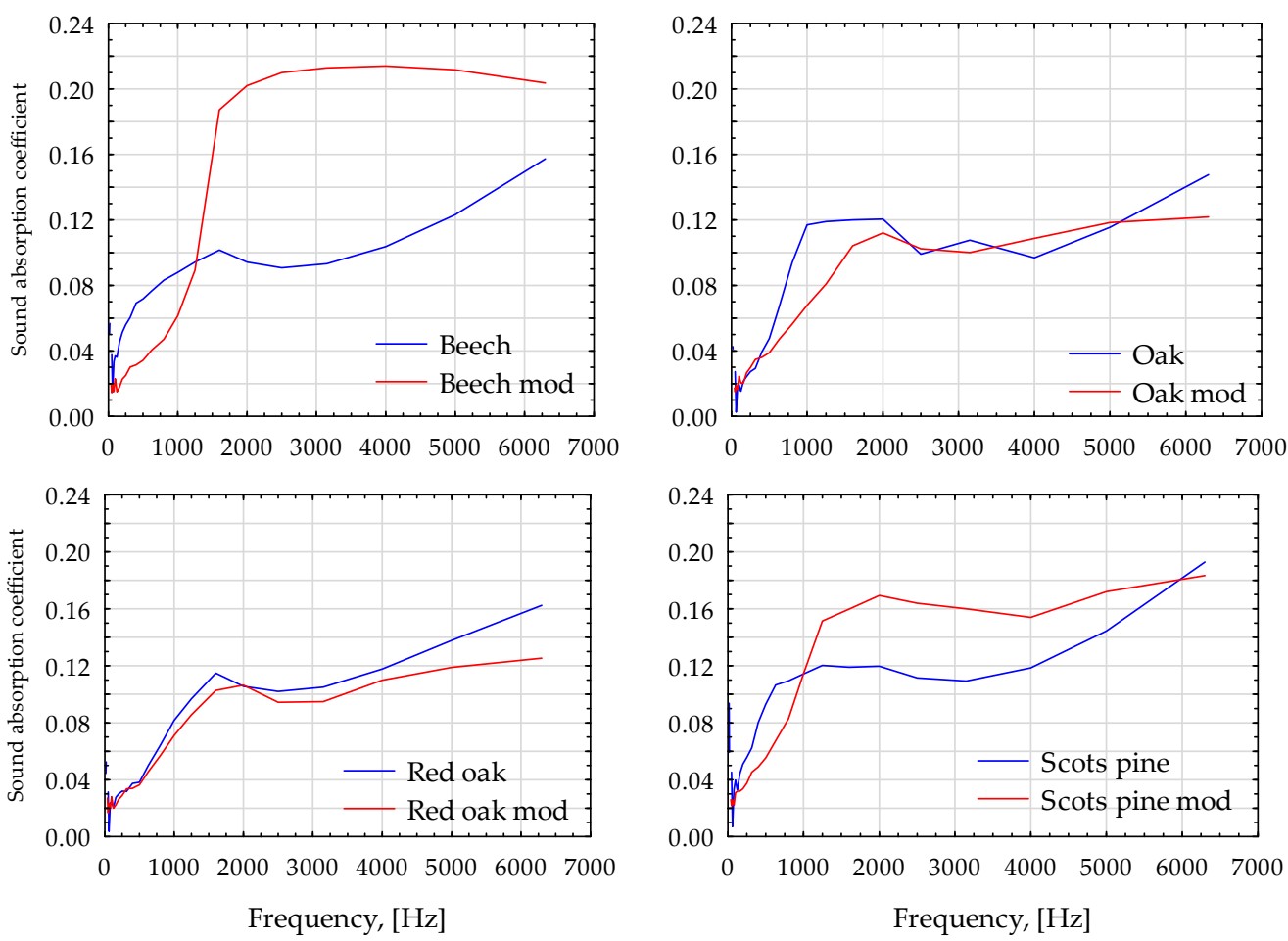

**Figure 6.** Comparison of the wave absorption coefficient in the analysed wood before and after thermal modification.

## 4. Conclusions

Wood is used as a sound-absorbing material in the form of acoustic panels for interior design and acoustic screens on highways. The thermal modification process of wood contributes to the loss of wood mass and, thus, its density. As a result of high-temperature exposure to wood, the speed of sound propagation and the sonic modulus of elasticity may increase. The most important factor, however, is its impact on the sound absorption coefficient. An increase in wood porosity suggests an increase in this parameter. However, during the tests, no significant effect of the modification on the sound absorption coefficient in oak and red oak wood was found. The reason for this may be the large pores (vessels) in the early wood, which contribute to the uniformity of this parameter throughout the tissue.

**Author Contributions:** Conceptualisation, P.M.; methodology, P.M.; performing the experiments, P.M., A.F., and M.P.; writing—original draft preparation, P.M., M.P., and A.F.; writing—review and editing, P.M.; supervision, P.M. All authors have read and agreed to the published version of the manuscript.

**Funding:** This research received no external funding.

**Institutional Review Board Statement:** Not applicable.

**Informed Consent Statement:** Not applicable.

**Data Availability Statement:** Data are available upon request to the corresponding author.

**Conflicts of Interest:** The authors declare no conflict of interest.

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
