# Peer review of "Sound Wave Absorption Coefficient and Sound Velocity in Thermally Modified Wood"

_applsci, doi:10.3390/app13148136_

Round 1

Reviewer 1 Report

1.       Please comment on the data of materials density in the Table 1 presented in kg/m-3, why in -3 degree?

2.       Why were these particular wood species chosen for the research when it is likely that neither oak or red oak will ever be used for road acoustic walls due to their high cost?

3.       Explanations of markings are missing, for example VRR; VLL; VTT...see Table 2.

4.       For what reason does the thermal treatment of wood increase its porosity?

5.       How did you determine the porosity, is it only by calculation or are there other methods to do it?

6.       Why was a sufficiently high frequency Hz chosen for research, even though the object of research will be used on roads. On roads, high frequency is completely irrelevant, because in cars, noise is generated by the engine, transmission and exhaust system. Research in various countries has proven [6] that the main source of low-frequency noise in a car is the exhaust system. (Ouis D. Annoyance caused by exposure to road traffic noise). In your case, the most relevant place is up to 200 Hz. Tell me for what purpose did you conduct research up to 5000 Hz in terms of acoustic walls?

7.       The passage of noise through the noise barrier or its absorption largely depends on: the material of the noise barrier (mass load), the angle of incidence of the sound wave and the sound frequency. Did the studies measure the angle of incidence of the sound wave?

8.       Based on the international standard EN ISO 11654, sound absorption classes are determined. This is especially relevant on roads, as walls are designed according to classes. What are the wood absorption classes of your choice?

9.       Page 6 the abbreviations Tab. And Table, please unify the terms.

110. A noise level of 62-65 dB(A) is said to be enough to increase the heart rate, which is very important when it comes to vehicle noise. Can you provide data on what the sound pressure level in dB would be in your case?

211. Most of the literature sources are older than 10 years. It really needs to be updated.

312. Clarify the presented conclusions, because they are very abstract.

Author Response

The authors sincerely appreciate the Reviewer's work and are very grateful for the professional and pointing comments and suggestions. 

  1.       Please comment on the data of materials density in the Table 1 presented in kg/m-3, why in -3 degree?

Table 1 shows the density in kg×m-3, not kg/m3. Hence, , there is a product with a negative cube instead of the quotient.

2.       Why were these particular wood species chosen for the research when it is likely that neither oak or red oak will ever be used for road acoustic walls due to their high cost?

All the species used in the research are very popular materials in Poland. They are gaining popularity every year. Determination of the sound absorption coefficient and the impact of thermal modification may contribute to increasing the potential applications of these species, in particular red oak.

  1. Explanations of markings are missing, for example VRR; VLL; VTT...see Table 2.

Explanation of symbols added to table description.

4.       For what reason does the thermal treatment of wood increase its porosity?

Thermal modification lowers the value of wood density. The decrease in density is mainly due to the loss of basic wood components, e.g. hemicelluloses and partly cellulose. This loss contributes to an increase in the porosity of the wood.

5.       How did you determine the porosity, is it only by calculation or are there other methods to do it?

In the work, the porosity was only calculated on the basis of the known density. It was not determined by known methods for example with use of porometer.

6.       Why was a sufficiently high frequency Hz chosen for research, even though the object of research will be used on roads. On roads, high frequency is completely irrelevant, because in cars, noise is generated by the engine, transmission and exhaust system. Research in various countries has proven [6] that the main source of low-frequency noise in a car is the exhaust system. (Ouis D. Annoyance caused by exposure to road traffic noise). In your case, the most relevant place is up to 200 Hz. Tell me for what purpose did you conduct research up to 5000 Hz in terms of acoustic walls?

Noise from car engines, exhaust systems, etc. is not discussed in this article.The tests were performed in accordance with the ISO 10534-2 standard, which assumes such a wide frequency range.

7.       The passage of noise through the noise barrier or its absorption largely depends on: the material of the noise barrier (mass load), the angle of incidence of the sound wave and the sound frequency. Did the studies measure the angle of incidence of the sound wave?

The angle of incidence of the sound wave on the sample was not taken into account.The tests were carried out by the ISO 10534-2 standard, i.e. in an impedance tube. This method is based solely on the perpendicular incidence of the wave on the tested sample. The angle of incidence is taken into account only in tests in a reverberation chamber by ISO 354. Still, for this method, 10 to 12 m^2 of the tested material should be used, which is why it was decided to try in an impedance tube, where samples with diameters of 100 and 29 mm are sufficient.

8.       Based on the international standard EN ISO 11654, sound absorption classes are determined. This is especially relevant on roads, as walls are designed according to classes. What are the wood absorption classes of your choice?

The tests were carried out in accordance with the ISO 10534-2 standard, i.e. in an impedance tube, and not according to ISO 354. Determination of sound absorption classes in accordance with ISO 11654 is possible only based on measurements of the reverberation coefficient of sound absorption determined in accordance with ISO 354. Therefore, this work does not sound absorption classes were determined.

  1. Page 6 the abbreviations Tab. And Table, please unify the terms.

There is one form of Table everywhere in the text.

 A noise level of 62-65 dB(A) is said to be enough to increase the heart rate, which is very important when it comes to vehicle noise. Can you provide data on what the sound pressure level in dB would be in your case?

The article did not deal with the subject of vehicle noise.

  1. Most of the literature sources are older than 10 years. It really needs to be updated.

The determination of the sound absorption coefficient is not a new topic. Few people in history have dealt with this phenomenon. Few also determined the influence of thermal modification on the values of this parameter.

  1. Clarify the presented conclusions, because they are very abstract.

Conclusions have been redrafted.

All changes in the text have been marked with a different font colour.

Reviewer 2 Report

The paper is focused on the characterization of thermal modified wooden samples from different taxon. In particular, the authors determined the sound wave absorption coefficient and sound velocity of the woods. The topic falls within the scope of the journal. The presentation and discussion of the results could be improved. On this basis, I recommend the publication after the following revisions:

-          Equations should be numbered.

-          Table 1. Errors on the delta density values (%) should be added.

-          Similarly to the previous comment, errors on the delta values (%) should be added.

-          It would be interesting to present some optical images of the characterized woods before and after the thermal treatment.

-          Introduction could be updated by quoting recent studies [Cellulose 2019, 26(16), pp. 8853-8865; ACS Applied Materials and Interfaces 2021, 13(1), pp. 1651-1661] on the characterization of wooden samples from different taxon. These studies evidenced that the taxon affects different physico-chemical properties, including thermal and mechanical resistance.

Minor editing of English language are required

Author Response

The authors sincerely appreciate the Reviewer's work and are very grateful for the professional and pointing comments and suggestions. 

  •          Equations should be numbered.

We added numbers to equations.

  •          Table 1. Errors on the delta density values (%) should be added.

The values were recalculated and the standard deviation for this quantity was calculated.

  •          Similarly to the previous comment, errors on the delta values (%) should be added.

The values were recalculated and the standard deviation for this quantity was calculated.

  •          It would be interesting to present some optical images of the characterized woods before and after the thermal treatment.

The diagram showing the samples and their sizes has been replaced with photos of the samples before and after modification.

  •          Introduction could be updated by quoting recent studies [Cellulose 2019, 26(16), pp. 8853-8865; ACS Applied Materials and Interfaces 2021, 13(1), pp. 1651-1661] on the characterization of wooden samples from different taxon. These studies evidenced that the taxon affects different physico-chemical properties, including thermal and mechanical resistance.

The following sentence has been added in the introduction: In addition to this commonly used method, you can also find others, e.g. regarding changes in the properties of archaeological wood [7,8].

All changes in the text have been marked with a different font colour.

Reviewer 3 Report

Dear authors,

The manuscript provides a potentially interesting investigation of the sound wave absorption coefficient and sound velocity in thermally modified wood. It would be better to make  major revisions and check the results of experiments.

1.       For better understanding and comparative analysis, it would be better to add the values of moisture content of specimens.

2.       The data in Tables 1 and 4 contradict each other.

3.       It is necessary to check the data in Table 4. How it is possible to explain the high value of MOE for pine wood compared with beech and oak wood.

4. What is the reason for the increase of stiffness for red oak after heat treatment?

Author Response

The authors sincerely appreciate the Reviewer's work and are very grateful for the professional and pointing comments and suggestions. 

  1.       For better understanding and comparative analysis, it would be better to add the values of moisture content of specimens.

We add a new paragraph in Chapter 2: 

"All determinations were made on wood with the same moisture content. Control samples were seasoned at 20°C and 40% air humidity. To obtain the same equilibrium moisture content of the modified samples, they were placed in a chamber with the same temperature with an air humidity of 65%. This allowed us to obtain the equilibrium moisture content of wood at 7%."

2.       The data in Tables 1 and 4 contradict each other.

Is it about the positive correlation of density with a modulus of elasticity? If so, the dynamic (sonic) modulus of elasticity mainly depends on the sound propagation speed in the wood. The speed was very high for pine wood resulting in high MOE values.

3.       It is necessary to check the data in Table 4. How it is possible to explain the high value of MOE for pine wood compared with beech and oak wood.

The dynamic (sonic) modulus of elasticity is calculated from formula no. 2. Its values, therefore, depend mainly on the value of the speed of sound propagation. High values of sound propagation speed characterise Pine. The static (linear) MOE is lower in pine than in beech or oak. 

4. What is the reason for the increase of stiffness for red oak after heat treatment?

As mentioned above, I do not clearly state that, in this case, the stiffness of the wood increased. Although during an adequately conducted thermal treatment process, it may increase (e.g. Optimization of Spruce (Picea Abies L.) Wood Thermal Treatment Parameters to Improve Its Acoustic Properties). The values of some acoustic parameters may increase because they depend to a large extent on the value of the speed of sound propagation.

All changes in the text have been marked with a different font colour.

Reviewer 4 Report

The paper discusses the sound absorption coefficient of wooden panels of different types of trees.

Smooth wood is not a sound-absorbing material in the classic sense, being rigid and not very porous, it has a low absorption coefficient value.

If the material is smooth, the absorption coefficient value is very low and therefore there is a high coefficient of sound reflection (r=1).

The values of the acoustic absorption coefficient by impedance tube of beech, oak, red oak and pine wood before and after thermal modification are reported.

Figures 4 and Figure 5 show the values as a function of frequency.

We have only four materials whose maximum absorption coefficient is 0.22. This value is very low to justify wood as a sound absorbing material.

I performed a test of the measurement of the sound absorption coefficient with the impedance tube with the hard surface instead of the specimen, to verify the value of the absorption coefficient in the condition of maximum reflection and compare this data with those obtained from the wood.

I would see if there are differences, being the measured coefficient very low I would check this condition.

Author Response

The authors sincerely appreciate the Reviewer's work and are very grateful for the professional and pointing comments and suggestions. 

A low sound absorption coefficient characterises wood. This coefficient usually oscillates in the range of 0.15-0.25, making wood a material in the absorption class E (according to PN-EN ISO 11654). However, wood is commonly used as a material whose task is to absorb and diffuse sound.

All changes in the text have been marked with a different font colour.

Round 2

Reviewer 1 Report

Thanks to the authors for the corrections. Many of the comments have been duly addressed. However, we would like the literature review not only to be supplemented by two sources, but also to have the rest updated and will be not older than 10 years.

Author Response

Many thanks to the reviewer for his time.

The list of literature has been supplemented with several other items.

Reviewer 2 Report

The paper was correctly revised by the authors. In my opinion, the paper can be accepted after the following minor revision:

- Ref. 8 should be checked. In the Bibliography, "2020" is wrongly indicated as year of the publication. The year of publication is "2021".

Minor editing of English language required

Author Response

Many thanks to the reviewer for his time.

All citations are taken from Google Scholar via Zotero. The authors did not interfere with the publication date of the article. However, it has been checked and corrected.

Reviewer 3 Report

Dear authors,

1.       For comparative analysis it would be better to add some papers about the influence of heat treatment on wood stiffness. You can use them to explain your results.

For example, from Borůvka, V.; Zeidler, A.; Holeček, T.; Dudík, R. Forests 2018, 9, 197

“There is a partial increase in the values of most properties at a lower treatment temperature, eventually leading to the preservation of values at the level of untreated wood, for example, for birch, the modulus of rupture increased by 26%, the modulus of elasticity by 24%, and the hardness in the radial plane by 34%. This is related to the fact that chemical changes are not yet significant, and they only case the restriction of the wood’s ability to absorb bound water.”

Borůvka, V., Zeidler, A., and Holeček, T. (2015). "Comparison of stiffness and strength properties of untreated and heat-treated wood of Douglas fir and alder," BioRes. 10(4), 8281-8294.

https://bioresources.cnr.ncsu.edu/resources/comparison-of-stiffness-and-strength-properties-of-untreated-and-heat-treated-wood-of-douglas-fir-and-alder/

Borůvka, V.; Zeidler, A.; Holeček, T.; Dudík, R. Elastic and Strength Properties of Heat-Treated Beech and Birch Wood. Forests 2018, 9, 197. https://doi.org/10.3390/f9040197

https://www.mdpi.com/1999-4907/9/4/197/htm

Author Response

(The authors gave the same response as above.)

Reviewer 4 Report

unfortunately you have made only a few changes to your paper;

you have not replied to my comments.

Author Response

Many thanks to the reviewer for his time.

The authors responded to all comments and suggestions. The authors needed to learn how to respond to the reviewer's comments, because they were rather informative and descriptive. They did not suggest changes but only emphasised the information contained in the text.

Changes and corrections have appeared in the text, marked with a different font colour.

Round 3

Reviewer 4 Report

the value of the absorption coefficient measured with impedance tube passes from a first measurement from 0.18 to 0.22

the value is little appreciable and significant.

You need to take an impedance measurement with a hard surface as a termination and see the difference from your measurements.

In my opinion, a gain of a few decimals does not justify the tests that have been performed.

 Extensive editing of English language required

Author Response

The authors sincerely appreciate the Reviewer's work and are very grateful for the professional and pointing comments and suggestions. 

  • the value of the absorption coefficient measured with impedance tube passes from a first measurement from 0.18 to 0.22

We obtained results that range from 0 to a maximum of about 0.22. However, these are results that are characteristic of solid wood. Wood with a density range of 450-850 kg/m3, i.e. the wood used in these tests, will not be a sound-absorbing material with higher sound absorption coefficients. However, the thermal modification of wood contributed to changes in this parameter. With the production of sound-absorbing material from wood, we will achieve higher coefficient values, as, for example, in fibreboards.

  • the value is little appreciable and significant.

The obtained values do not change significantly; however, the main reason is that the maximum value of the analysed coefficient is 1. For solid wood, it rarely exceeds 0.3. In beech and pine wood, we observed clear changes in the value of this parameter, which may contribute to a more conscious choice of modified wood for the production of sound-absorbing materials.

  • you need to take an impedance measurement with a hard surface as a termination and see the difference from your measurements.

It should be clarified here what the reviewer means by the term "hard surface". Wood is a compact material with relatively high hardness. Lower than other materials (steel, concrete). However, if we used a very smooth, low-porous surface for testing, the obtained results would be even lower, close to 0.

  • In my opinion, a gain of a few decimals does not justify the tests that have been performed.

The obtained results differing in decimals are due to the low value of the analysed parameter. It reaches a maximum value of 1. Therefore, a change of one-tenth means a change of the parameter by 10%. In this system, the decimal difference is significant. In addition, thermal modification reduces the hygroscopicity of wood and contributes to increasing the durability of wood.